# Composite Proton-Conducting Membrane with Enhanced Phosphoric Acid Doping of Basic Films Radiochemically Grafted with Binary Vinyl Heterocyclic Monomer Mixtures

**DOI:** 10.3390/membranes13010105

**Published:** 2023-01-13

**Authors:** Paveswari Sithambaranathan, Mohamed Mahmoud Nasef, Arshad Ahmad, Amin Abbasi, T. M. Ting

**Affiliations:** 1Centre of Hydrogen Energy, Institute of Future Energy, Universiti Teknologi Malaysia, Jalan Sultan Yahya Petra, Kuala Lumpur 54100, Malaysia; 2Malaysia-Japan International Institute of Technology, Universiti Teknologi Malaysia, Jalan Sultan Yahya Petra, Kuala Lumpur 54100, Malaysia; 3Chemical Engineering Department, Universiti Teknologi PETRONAS, Seri Iskandar 32610, Perak, Malaysia; 4Radiation Processing Technology Division, Malaysian Nuclear Agency, Kajang 43000, Selangor, Malaysia

**Keywords:** proton conducting membrane, radiation grafting on EB irradiated film, reactivity ratio of 4-VP and 1-VIm, enhancing acid doping, proton conductivity, HT-PEMFC performance

## Abstract

A composite proton conducting membrane (PCM) was prepared by radiation-induced grafting (RIG) of binary mixtures of 4-vinyl pyridine (4-VP) and 1-vinylimidazole (1-VIm) onto poly(ethylene-*co*-tetrafluoroethylene) (ETFE) film followed by phosphoric acid (PA) doping. The grafting parameters such as absorbed dose, temperature, monomer concentration, time, and monomer ratio were varied to control the degree of grafting (DG%). The effect of the reactivity ratio of 4-VP and 1-VIm on the composition and degree of monomer unit alternation in the formed graft copolymer was investigated. The changes in the chemical and physical properties endowed by grafting and subsequent PA acid doping were monitored using analytical instruments. The mechanical properties and proton conductivity of the obtained membrane were evaluated and its performance was tested in H_2_/O_2_ fuel cell at 120 °C under anhydrous and partially wet conditions. The acid doping level was affected by the treatment parameters and enhanced by increasing DG. The proton conductivity was boosted by incorporating the combination of pyridine and imidazole rings originating from the formed basic graft copolymer of 4-VP/1-VIm dominated by 4-VP units in the structure. The proton conductivity showed a strong dependence on the temperature. The membrane demonstrated superior properties compared to its counterpart obtained by grafting 4-VP alone. The membrane also showed a strong potential for application in proton exchange membrane fuel cells (PEMFC) operating at 120 °C.

## 1. Introduction

Functional polymeric materials play a very crucial role in driving progress in the surging quest to acquire renewable energy systems based on electrochemical reactions [1]. Particularly, these functional materials are widely investigated to develop polymer electrolytes and separators for reliable and sustainable energy conversion and storage devices such as supercapacitors, batteries, electrolyzers, and fuel cells [2]. Of all, proton conducting membranes (PCMs) play an essential role in proton exchange membrane fuel cells (PEMFCs), which convert the free energy of H_2_ fuel in the presence of O_2_ into electricity using an electrochemical catalyst with water as a by-product. To allow the flow of electrons in the outside circuit, PCMs (such as Nafion) parallelly transport protons from the anode to the cathode while preventing the bulk mixing of ionized O_2_ and H_2_. The proton transport takes place in the aqueous phase of the membrane, which swells, forming inter-connected ionic clusters for proton-hopping and diffusion. Thus, the membrane water content must be controlled by operating PEMFC at temperatures ≤ 80 °C [3]. However, low temperature (LT) PEMFC operation is challenged by the need to use pure H_2_ to prevent catalyst poisoning by CO and the system complexity caused by using water and heat-management systems. Therefore, it is highly desirable to operate PEMFC above 100 °C and this requires less water-dependent membranes that differ from those containing terminal sulfonate groups on a perfluorinated side chain attached to a Teflon backbone such as Nafion [4]. The latter and its developmental counterparts undergo a loss of humidity which causes a phase separation between hydrophilic and hydrophobic domains in the membrane, leading to an insufficient water uptake content to provide the proton transport channels leading to proton conductivity reduction that is accompanied by a decline in the electrochemical stability, and mechanical strength, when operated above 100 °C [5,6].

The interest in the development of PCMs with high conductivity and durability for high-temperature (HT) PEMFC is receiving increasing attention [7]. This is to increase CO tolerance, allow the use of reformed gases, enhance the cathode kinetics, and reduce the system complexity by eliminating the water management system [8]. This motivated researchers to develop a wide range of alternative membranes such as Nafion composites, sulfonated partially fluorinated and non-fluorinated hydrocarbon acid-doped polymers, inorganic/organic nano-hybrids, solid acids with super-protonic phase transition membranes, acid/base membranes, polybenzimidazole (PBI), pyridine-substituted PBI doped phosphoric acid (PA) composites, carbon-based PBI doped membranes, and metal-organic framework-containing membranes. More details on these membranes, including their preparation methods, properties, proton conduction mechanisms, and state of their application of HT-PEMFC, can be found in a large number of published reviews [7,9,10,11,12,13,14,15,16,17,18,19,20,21].

Various preparation methods have been used for the development taking place in such proton conducting membranes. Of all, RIG is an appealing method that is well-known for covalently imparting side chain grafts originated from parent polar monomers to preformed polymer films. This technique provides a compelling way to prepare ionic membranes by integrating many vinyl monomers with polymeric sheets/films as starting substrates to acquire functional groups imparting desired properties. Thus, RIG has been widely utilized to prepare a variety of polymer electrolyte membranes for various types of electrochemical storage including vanadium redox flow batteries and electrochemical conversion devices, such as PEMFC [22,23,24,25,26,27,28]. Of particular interest, PCMs for HT-PEMFC were pursued several years ago, and few alternative acid/base membranes were prepared by RIG of 4-vinylpyridine (4-VP) or 1-vinylimdazole (1-VIm) onto partially fluorinated polymers such as poly(vinylidene fluoride) (PVDF) and poly(ethylene-*co*-tetrafluoroethylene) (ETFE) films followed by PA acid doping under controlled conditions. These membranes, which have H-bonds network of pyridine or/and imidazole rings and phosphoric acid gives sites enhancing proton hopping and reducing acid leaching showed a potential to replace PBI/PA membranes [29,30,31]. Such membranes were reported to have relatively low conductivity (9.6 mS/cm) and to be prone to acid leaching, lowering their conductivity when used in HT-PEMFC [32].

The method of preparation of radiation-grafted PCMs starts with a grafting step involving vinyl heterocyclic monomers, which is essential to form basic precursor films capable of hosting PA at a loading level that is a function of the degree of grafting (DG%), PA concentration, reaction time, and temperature [29,30,31]. The obtained membranes demonstrated an improved proton conductivity in the range of 20–30 mS/cm at 120 °C compared to the PBI/PA membrane. However, such values are lower than that of the Nafion membrane when tested above 80 °C (85 mS/cm) [33]. Moreover, the performance of the membranes was found to be rather unsatisfactory and reports addressing their acid-leaching resistance and stability were not published. To further improve these membranes, RIG of comonomers (adding a second monomer to the main grafting monomer) was used to enhance the acid doping level by incorporating more basic characters and improving the stability of the membrane [34,35]. However, grafting binary monomer mixtures with a basic nature, such as 4-VP and 1-Vim, is challenging and requires understanding key issues, such as the effect of monomer reactivity ratio and its impact on the copolymer composition relative to the monomer mixture content and the sequence of alternation of individual monomer units in the copolymer, together with its impact on the acid doping level in the final composite membranes.

The objective of the present study is to report the preparation of PCM with enhanced acid doping by RIG of 4-VP and 1-VIm binary mixtures under manipulated reaction parameters followed by PA immobilization. The monomer reactivity ratio and run number were investigated and correlated to the structure developed in the copolymer film relative to monomer units in the bulk grafting solution and the properties of the obtained membrane. The membrane’s various physical and chemical properties were evaluated, and its performance was tested in HT-PEMFC under controlled dynamic conditions.

The present investigation is essential to allow quantitative prediction of 4-VP/1-VIm relative reactivity towards the graft growing chains and the formed composition in the graft copolymer compared to their initial comonomer ratio in the grafting solution. This would assist in preventing random polymerization in the grafted film and enable tailoring the membrane architecture. Moreover, the introduction of a combination of imidazole ring with a pyridine ring is expected to bring a stability improvement by letting the N atom of the latter react with hydroxyl radicals generated in the PEMFC medium forming pyridine-N-oxide, which is likely to impede the oxidation reactions of imidazole groups as reported elsewhere [36].

## 2. Materials and Methods

### 2.1. Materials

ETFE films with a thickness of 50 μm and a density of 1.7 g/cm^3^ were obtained from Goodfellow Cambridge Ltd. (Cambridge, UK). 1-VIm (purity ≥ 99%, Sigma-Aldrich, St. Louis, MO, USA) and 4-VP (purity > 95%, ACROS) were used as received without further purification. PA with 85% concentration was purchased from Mallinckrodt Chemicals (Thailand). Deionized water (DI) that was used in the experiments was produced using a NANOpure1 DIamond™ water purifier (Taylor Scientific., St. Louis, MO, USA).

### 2.2. Membrane Preparation

The membrane preparation was carried out using 3 sequential steps including irradiation, graft copolymerization, and PA doping. The ETFE films of the desired size were initially cleaned with ethanol and kept in polyethylene (PE) bags before they were evacuated and thermally sealed. The samples were irradiated with an electron beam (EB) to total doses in the range of 20–100 kGy at a10 kGy/pass under ambient temperature using an NHV-Nissin High Voltage, EPS 3000 accelerator (Nissin, Japan) operated at a voltage of 2 MeV and a beam current of 2 mA. The EB irradiated films in the sealed PE bags were kept in a low-temperature freezer (−65 °C) for a day before they were used.

The grafting step was carried out by placing the irradiated samples in an evacuated ampoule containing an oxygen-free grafting solution composed of desired ratios of 4-VP/1-VIm mixtures diluted with DI. The process of sample evacuation and deoxygenation of the grafting solution by bubbling with pure N_2_ was carried out using a grafting apparatus that was described elsewhere [37]. All the ampoules containing grafting mixtures were placed in a thermostatic water bath under a controlled temperature and the grafting reaction was performed for the intended period. The grafted films were removed, washed with 0.1 M HCl solution, and rinsed therein under sonication for 16 h. The obtained membrane precursors were dried under vacuum at 60 °C for 24 h. The DG% was calculated according to Equation (1):(1)$$DG\left(\%\right)=\frac{W_{g}-W_{o}}{W_{o}}\times 100$$
where, *W_o_* and *W_g_* are the weights of ETFE film before and after grafting, respectively.

The grafted films were functionalized by doping with PA at various reaction parameters in different ranges including 4-VP/1-Vim ratio, PA concentration, temperature, and reaction time. After the reaction was completed, the samples were abstracted, and the excess acid was removed followed by keeping them at 80 °C in a vacuum oven overnight. The doping level was determined according to Equation (2):(2)$$DL(\%)=\frac{W_{d}-W_{g}}{W_{g}}\times 100$$
where, *W_d_* is the weight of the membrane after doping with PA.

The membranes grafted with binary monomer mixtures and those grafted with individual monomers that were subsequently doped with PA were denoted as ETFE-*g*-P(4-VP/1-VIm)/PA, ETFE-*g*-P(4-VP)/PA and ETFE-*g*-P(VIm)/PA, respectively. The intermediate grafted films were designated as ETFE-*g*-P(4-VP/1-VIm), ETFE-*g*-P(4-VP), and ETFE-*g*-P(VIm).

### 2.3. Membrane Characterization

FTIR measurements were performed on PA-doped membranes in comparison with pristine ETFE film and grafted samples using a Perkin Elmer-Spectrometer (2000 Explorer) in a transmission mode in the frequency range of 500–4000 cm^−1^ at a scanning rate of 16 with a resolution of 4 cm^−1^.

X-ray diffraction (XRD) measurement was performed using an X’pert Pro X-Ray diffractometer model PW 3040 Philips. The diffractograms were measured employing Nickel-filtered Cu-Kα radiation (λ = 1.541 Ǻ) and data were obtained at a scanning range of 2θ of 5–50°.

Thermal gravimetric analysis (TGA) of the membrane was carried out by a Pyris-1 TGA at a constant heating rate of 20 °C/min in a temperature range of 50–700 °C under N_2_ atmosphere at 100 mL/min flow rate.

The mechanical properties of the samples were investigated using a Shimadzu AG-X plus universal mechanical tester according to ASTM-D882 at a crosshead speed of 5 mm/min at room temperature.

Elemental analysis was carried out using LECO CHN628 analyzer to determine the content of C, H, and N in the grafted samples. The obtained data were used to estimate the reactivity ratio of 4-VP and 1-VIm during RIG of their mixtures onto ETFE films. Particularly, the 4-VP mole fraction in the comonomer solution mixture was monitored and compared with the counterpart fraction in the copolymer (grafted film in this case) which was calculated using the equation described by Bakhshi et al. [38].

Proton conductivity of the PA-doped membranes was measured by the four-probe conductivity cell (Bekk. Tech. Inc., Katy, TX, USA) connected to a DC source meter (Keithley 2400, Keithley Instruments Inc., Cleveland, OH, USA) and controlled by a Lab view software. The membrane strips with dimensions of 5 mm × 25 mm were sandwiched between Pt electrodes placed in the cell Teflon cell as detailed elsewhere [39]. The activation energy (*E_a_*) in J/mol was calculated using the Arrhenius equation given below:log σ = log σ_0_ − (*E_a_*/RT) (3)
where σ is the proton conductivity (mS/cm) and σ_0_ is the pre-exponential factor. R is the universal gas constant (8.314 J/mol K), and T is the absolute temperature in K.

The chemical stability of the membrane was evaluated in a simulated oxidative environment using the Fenton test as reported elsewhere [40]. The weight loss of samples was monitored after immersion into 50 mL Fenton solution comprising 3 wt% H_2_O_2_ and 4 ppm Fe^2+^ from ferrous sulfate at 68 °C. The sample weight was measured every 12 h after removal and washing a few times in DI. This was followed by drying at 80 °C for 16 h in a vacuum oven before immediate weight recording. Fresh Fenton reagent was used in every immersion cycle.

### 2.4. Fuel Cell Test

The fuel cell performance of ETFE-*g*-P(4-VP-*co*-1-VIm)/PA membrane was tested in a single cell with an active area of 5 cm^2^ using a computer-controlled fuel cell test station (Scribner 850E). The membrane electrode assemblies (MEAs) were made using ELAT single sided coatings type carbon cloth electrodes from E-TEK (Somerset, NJ, USA) with a Pt loading of 0.4 mg/cm^2^ (20% Pt/Vulcan XC72). The membrane, which was first dipped in PA (85 wt%) for a few minutes and gently plotted with polypropylene (PP) tissue sheet, was sandwiched between the electrodes and hot pressed at 120 °C and 5 MPa for 30 s to fabricate the MEA. The cell was operated under anhydrous conditions or 20% relative humidity (RH%) and flow rates of 450 for H_2_ and 300 mL/min for O_2_ at atmospheric pressure with no back pressure detected. The current-voltage characteristics were recorded using multi-range programmable electronic load interfaced with Fuel Cell (Windows 2010/XP) software.

## 3. Results and Discussion

### 3.1. Effect of Grafting Parameters on Degree of Grafting

The effects of reaction parameters such as absorbed dose, monomer concentration, temperature, and reaction time on the DG (the content of 4-VP and 1-VIm copolymer) in the grafted films were investigated and the obtained data are presented in Figure 1. The increase in the absorbed dose led to a gradual increase in DG as shown in Figure 1A. This was caused by the rise in the number of radicals formed in the irradiated ETFE film that enhanced the initiation reaction [41].

The rise in the monomer concentration depicted in Figure 1B caused an increase in the DG until a concentration of 60 vol%, because of the abundance of monomer molecules at the grafting sites caused by diffusion enhancement beyond which DG decreased gradually. This behavior is ascribed to the increase in the viscosity in the grafting zone caused by the partial homopolymerization which hampered the monomer diffusion and reduced its availability in the grafting sites leading to lesser chain propagation. A similar trend was reported for RIGC of styrene onto PP films [42].

The DG was found to be a function of temperature as indicated in Figure 1C. For instance, the DG sharply increased with the increase in the temperature until reaching a maximum value at 60 °C, beyond which it remarkably decreased. This behavior can be attributed not only to the increase in the trapped radical activation but also to the enhancement of the monomer diffusion to the grafting sites leading to faster kinetics i.e., enhancing rates of initiation and propagation. Higher temperatures beyond 60 °C led to the accumulation of excess monomers in the grafting sites triggering homopolymerization that hindered the monomer accessibility to the grafting sites. This was accompanied by the decay of primary radicals and termination of graft-growing chains by mutual combination or bimolecular termination. Similar temperature-dependent behavior for DG was reported for grafting of vinylbenzyl chloride on ETFE films by Wange et al. [43].

The variation of the DG with the reaction time illustrated in Figure 1D showed that the DG increases with time up to 18 h, beyond which it reached saturation. It is obvious that the rate of grafting increased steadily in the first 10 h and this is due to the slow diffusion of monomers through the film surface layers leading to lower DG indicating the grafting occurs at the surface only. However, a further increase in the reaction time led to an increase in the DG because of more monomer diffusion and presence in the grafting sites. It can be also suggested that the surface grafted layers were swollen in the monomer solution, allowing the consecutive diffusion of monomer molecules to the inner grafting sites. The grafting saturation observed after 18 h is likely to be due to the meeting of the graft growing chains emerging from both film surfaces inward towards the middle of the film until achieving a complete volume grafting and leaving no room for more grafting, despite the prolonged time. Similar trends for the grafting of various comonomer systems were reported in the literature [30,44,45].

Figure 2 displays the variation of DG with the comonomer ratio for grafting of 4-VP/1-VIm mixtures onto ETFE films in the range of 20–80%. Pure individual monomers (100% 4-VP corresponding to 0% 1-VIm and vice versa) were used as references. It can be clearly seen that DG increased with the increase in the content of 1-VIm in the comonomer mixture that accompanied the decrease in 4-VP content reaching the highest value of 83% at a 4-VP/1-VIm composition of 50/50 vol%. This DG value is higher than the maximum amount of DG that could be obtained with grafting of individual monomers (4-VP or 1-VIm). When a binary monomer mixture is grafted, the molecules of the most reactive one diffuse faster inside the swollen film and subsequently carry the least reactive monomer molecules present in the solution. Therefore, the overall monomer concentration in the grafting zone increases leading to a complementary situation for higher graft copolymer formulation. Therefore, a synergistic effect in the grafting process of 4-VP/1-VIm binary mixture onto ETFE film took place. This trend goes along with the observation reported in the literature for RIG of acrylonitrile and acrylic acid binary mixture onto PE film [46]. However, the increase in the concentration of 1-VIm was observed to cause a decrease in DG. This could be due to the increase in the viscosity of the monomer mixture that hinders the propagation reaction [47] and/or steric hindrance caused by the presence of basic pyridine and imidazole rings in the grafting zone [35,48].

### 3.2. Reactivity of 4-VP and 1-VIm in Monomer Mixture

The effect of reactivity of 4-VP and 1-VIm during the RIGC of their mixtures onto ETFE films was investigated. The mass % of C, H, and N were obtained from the elemental analysis data of the copolymer and the mole fraction used in the comonomer solution was evaluated in comparison with that of the formed copolymer (grafted film), and the obtained data are presented in Figure 3. The monomer reactivity ratios for 4-VP and 1-VIm were determined according to Mayo–Lewis copolymerization [49] and were found to be 0.78 for 4-VP and 0.23 for 1-VIm. Moreover, the product of the reactivity ratio was found to equal 0.29, which is below the unity value representing the theoretical ideal copolymerization. This is proving that there is an obvious tendency to deviate from the ideal alteration of comonomer units in the copolymer chains and 4-VP units favorably dominated the copolymer formed in the grafted ETFE film. Moreover, the presence of 1-VIm decelerated the copolymerization of 4-VP. The reactivity ratios obtained for two monomers in this study agree with the literature for copolymerization of 4-VP and 1-VIm by free radical polymerization in bulk [50]. However, the present copolymerization system differs because the reaction takes place in the film in which factors such as the solvent used, base film swelling and the monomer diffusion, all of which affect the DG and the composition of the formed graft copolymer film [45]. Particularly, the monomer diffusion is affected by the steric hindrance present in heterocyclic monomers, having cationic N centers providing steric shielding, that are more profound in 1-VIm than 4-VP [51]. In other words, the strong steric hindrance in 1-VIm reduces its diffusion to the grafting sites compared to 4-VP. This trend highlights that graft copolymerization of the comonomer system onto ETFE films seems to be more complex compared to free radical polymerization of the same system in bulk solution.

### 3.3. Effect of Reactivity on Degree of Alteration in Formed Graft Copolymer

The product of the reactivity ratio of 4-VP and 1-VIm, which equals 0.29, indicates the presence of an alteration in the grafted films. The degree of alternation in the copolymer is determined by the “run number”, which is the measure of sequence distribution or the number of uninterrupted monomer sequences (A-A) or (B-B) in a copolymer chain per hundred monomer units. The run number in ETFE grafted films with 4-VP/1-VIm mixtures is determined from the 4-VP molar fraction in the grafting solution and the reactivity ratios according to Harwood [52] and the data is presented in Figure 4. The maximum value of the run number was obtained at a molar fraction of 0.6 at which the tendency to alternate in the copolymer chains is most likely taking place and beyond which the run number decreased. At this composition, ~65% of the grafted chains were in the form of 4-VP units followed by 1-VIm units. The deviation from the run number of 1 is most likely caused by the discrepancy in the monomer reactivity ratio in such heterocyclic monomers and other factors such as polarity, steric hindrance, and stabilization by the resonance of the radicals generated during the copolymerization process [53].

### 3.4. Enhancement of Acid Doping Level

Figure 5 shows the variation of acid doping level for P(4-VP/1-VIm)-grafted ETFE films at different treatment parameters. P(4-VP)-grafted ETFE film was used as a reference. The increase in PA concentration led to a drastic increase in DL (Figure 5a) with the maximum value achieved at the conditions of 59% DG and 85% PA for 24 h at 80°. On the other hand, the acid uptake was found to be remarkably higher in the grafted membranes containing monomer mixtures compared to the counterpart containing P(4-VP) alone. The incorporation of the two basic monomers enhanced the swelling and diffusion of acid significantly promoting the interaction (complexation) between both -N^+^- of the imidazole ring of P(1-VIm) and pyridine ring of P(4-VP) with PA at higher rates until all the imidazolium and pyridinium cations were completely reacted.

The increase in the doping temperature enhanced the rate of reaction and led to a gradual rise in DL when the temperature increased from 30 to 70 °C and continued to increase, but with a slower pace to 80 °C as shown in Figure 5b. The membrane grafted with a comonomer system (4-VP/1-VIm) showed a higher DL than the ETFE film grafted with 4-VP alone, confirming the enhancement of basic characters imparted by grafting this binary mixture.

A similar increasing trend was observed for DL with the elaboration of reaction time as depicted in Figure 5c, showing the variation of DL of P(4-VP/1-VIm)-grafted films with time. The DL reached a maximum value in the P(4-VP/1-VIm)/PA membrane after 4 days with DL at all treatment times, higher than those of the counterpart grafted with 4-VP alone. The best reaction parameters were identified and applied to investigate the variation of acid doping level with the 4-VP/1-VIm ratio in the grafting solution and the obtained data is presented in Figure 5d. As can be seen, a maximum DL of 119% was achieved for a sample having a DG of 59% treated with 85% PA and at 80 °C and 4 days. The presence of positive nitrogen centers of different strengths originating from the heterocyclic rings of both monomers synergized the PA-loading reaching the maximum value.

### 3.5. Changes in Properties

#### 3.5.1. Chemical Composition Changes

Figure 6 shows typical FTIR spectra of the ETFE-*g*-P(4VP/1-VIm)/PA membrane together with its grafted counterpart. The pristine ETFE and P(4-VP)-grafted films were used as references. The spectrum of ETFE (Figure 6a) displayed broad characteristic bands in the range of 1000–1400 cm^−1^ representing CF_2_ groups coupled with a small band at 2915 cm^−1^ for the stretching vibration of CH_2_ groups. The P(4-VP)-grafted control sample showed a pyridine ring representing peaks at 1600, 1560, 1460, and 1425 cm^−1^ assigned for C=C and C=N in addition to C-H peaks in the range of 725–880 cm^−1^. This confirms the successful grafting of 4-VP onto film-grafted ETFE film (Figure 6b). Similarly, the poly(4VP/1-VIm)-grafted film displayed several peaks in ranges of 1525–1575 and 725–880 cm^−1^, having lower intensity resulting from the surface domination with combinations of N–H and C–H in the range of 2250–3180 (Figure 6c). This was accompanied by the emergence of an OH broad peak in the range of 3270–3550 cm^−1^ caused by the formation of an H-bonding network between the imidazole/pyridine, imidazole/water, and pyridine/water incorporated in the grafted films. This imparted hydrophilic characteristics that are more prevailing in the film grafted with the comonomer mixture (4VP/1-VIm) than in that grafted with 4VP monomer. These observations confirm the successful grafting of 4VP/1-VIm mixture onto ETFE film. The doping of P(4VP/1-VIm) grafted film with PA formed broader peaks in the region of 2000–3500 cm^−1^, which was accompanied by the appearance of new peaks at 962, 885, and 810 cm^−1^ representing PA. Therefore, the mechanism depicted in Figure 7 can be suggested for the preparation of the PA-doped membrane by RIG of 4VP and 1-VIm and subsequent acid doping.

#### 3.5.2. Morphological Changes

Figure 8 shows SEM images of the cross-sectional view of a ETFE-*g*-P(4-VP-*co*-1-VIm)/PA membrane with low and high magnifications. The membrane seems to have a uniform thickness and homogenous structure. The EDX analysis revealed the presence of carbon (31.78 wt%), fluorine (17.63 wt%), nitrogen (4.0 wt%), oxygen (36.76 wt%), and phosphorus (9.83 wt%), representing the elemental composition of the membrane originated from in ETFE substrate and phosphoric acid-doped poly(4-VP/1-VIm) grafts. The phosphorus mapping across the membrane further confirmed the presence of a homogeneous and uniform distribution of PA-doped grafts in the membrane.

#### 3.5.3. Structural Changes

Figure 9 shows typical XRD diffractograms of ETFE-*g*-P(4-VP/1-VIm)/PA and ETFE-*g*-P(4-VP)/PA membranes. The pristine ETFE and counterparts grafted with 4-VP/1-VIm and 4-VP were used as references. Compared to ETFE film semi-crystalline structure marked by the peak at Bragg’s angle of 2θ = 19°, grafting of either 4-VP or 4-VP/1-VIm comonomer mixture led to a reduction in crystallinity peak intensity of ETFE without any significant alternation in the crystalline peaks. Further reduction in the crystalline peak was observed after acid doping, and this was more profound with the membrane grafted with 4-VP/1-VIm compared to counterpart grafting with 4-VP alone. This observation can be attributed directly to the reduction in the crystalline structure caused by the dilution with the incorporated amorphous copolymer grafts which was further increased after acid doping. The changes in the structure of the membranes in this study are in complete agreement with those reported in the literature for similar membranes grafted with 4-VP and doped with PA [30,31].

#### 3.5.4. Thermal Stability Changes

The TGA thermograms of pristine ETFE, P(4-VP), and P(4-VP/1-VIm)-grafted films and their acid-doped counterpart membranes are shown in Figure 10 and the thermal degradation data are presented in Table 1. The pristine ETFE film showed a single-step degradation pattern at 440 °C due to the decomposition of the polymer backbone. The P(4-VP)-grafted film demonstrated a two-step degradation pattern started by the depolymerization at 340 °C followed by decomposition of polymer backbone at 470 °C suggesting that the incorporation of P(4-VP) enhanced the stability of ETFE film. On the other hand, grafting of 4-VP/1-VIm led to emerging of a new degradation step in a form of prolonged dehydration in the range of 50–150 °C in P(4-VP/1-VIm) grafted film caused by the presence of H-bonds between water molecules and the basic pyridine and imidazole rings. This was followed by depolymerization at 290 °C and decomposition of ETFE backbone at 445 °C suggesting that grafting of 4-VP/1-VIm accelerated the decomposition of the grafted film components and imparted remarkable hydrophilic character as indicated by the continuous dehydration weight loss up to 150 °C caused by the presence of H-bonds. The PA doping for both grafted films inherited the membranes 4-step degradation pattern in response to heat treatment. For instance, ETFE-*g*-P(4-VP)/PA membrane displayed a dehydration step up to 150 °C, evaporation of PA at 200 °C, depolymerization at 340 °C and ETFE backbone decomposition at 485 °C. A similar degradation pattern is observed for P(4-VP/1-VIm)/PA membrane starting with the dehydration up to 150 °C, evaporation of PA at 200 °C, depolymerization at 320 °C, and ETFE backbone decomposition at 450 °C. This suggests that the doping of grafted films with PA enhanced the thermal stability of the grafted component and ETFE backbone. The thermal degradation behavior of these membranes is going along with that of radiation-grafted and PA-doped counterparts reported in the literature [34,35]. It can be concluded that the prepared PA doped membranes have a thermal stability up to 200 °C and are suitable for applications in HT-PEMFC below 160 °C.

#### 3.5.5. Mechanical Properties Changes

Figure 11 shows stress–strain diagrams of PA-doped ETFE-*g*-P(4-VP/1-VIm) membrane in comparison with the pristine and ETFE-*g*-P(4-VP/1-VIm) films. As can be seen, the grafting of the monomer mixture onto ETFE reduced the tensile strength of ETFE film, which was further reduced by PA acid doping. For instance, the ETFE film, which has a tensile strength of 37.4 MPa coupled with an elongation of 104%, showed a reduced tensile strength of 21.6 MPa and an elongation of 57.8% after being grafted with the comonomer mixture. These values were further reduced to 12.1 MPa and 51% after doping with PA, leaving reasonable mechanical integrity in the PA-doped membrane, higher than that of PA-doped PBI (3.4 MPa), which is suitable for fuel cell application.

#### 3.5.6. Proton Conductivity Changes with Temperature

Figure 12 shows the variation of proton conductivity with temperature for ETFE-*g*-P(4-VP/1-VIm)/PA compared to their corresponding PA-doped membranes, obtained from the grafting of individual 4-VP and 1-VIm monomers onto ETFE films and having the same DG (i.e., 59%). The proton conductivity was found to increase with the rise in temperature in all membranes while maintaining superior values in ETFE-*g*-P(4-VP/1-VIm)/PA compared to other membranes. For instance, the highest proton conductivity reached 75.4 mS/cm for ETFE-*g*-P(4-VP/1-VIm)/PA compared to 43.5 mS/cm for ETFE-*g*-P(1-VIm)/PA, and 33.4 mS/cm for ETFE-*g*-P(4-VP)/PA membranes. Since the proton conductivity of PA-doped membranes is known to take place by the alternate cycle of formation and dissociation of H-bonds network causing proton hopping [54], the proton conductivity rise in all membranes can be attributed to the enhancement of protons mobility (Brownian motion) and reduction in charge transfer resistance [55]. The superior proton conductivity values in ETFE-*g*-P(4-VP/1-VIm)/PA membrane are due to the increase in the number of proton nitrogen receptor sites originating from the incorporated pyridine and imidazole rings, which increased both acid DL and proton conductivity in turn. These results confirm that the grafting of 4-VP/1-VIm binary monomer mixture onto ETFE is more advantageous than grafting of individual monomers in imparting more basic characters to the copolymer films in forming more proton acceptor sites, allowing greater H-bonds network for proton mobility.

#### 3.5.7. Changes in Activation Energy

To elaborate on the electrochemical performance of PA doped membranes, Arrhenius curves were obtained by plotting proton conductivity vs. reciprocal of temperature using Equation 3, as shown in Figure 13. The *E_a_* of the membranes obtained from the slope of the linearly fitted relationships showed a similar trend, with minor differences in the *E_a_* values. For instance, the ETFE-*g*-P(4-VP/1-VIm)/PA membrane showed an *E_a_* of 5.21 kJ/mol compared to 5.5 kJ/mol for ETFE-*g*-P(4-VP)/PA. The lower *E_a_* in ETFE-*g*-P(4-VP/1-VIm)/PA membrane also confirms that the presence of more protonated N originated from imidazole and pyridine rings enhanced the acid doping level and boosted the proton conductivity. Particularly, the doping of poly(4-VP/VIm) grafted film with PA led to the formation of NH^+^ proton donor sites in the membranes causing H^+^ hopping to mainly take place between NH^+^ sites (from imidazole and pyridine rings) and PA anions (H_2_PO_4_^−^), leading to continuous proton transfer. It can be suggested that proton conductivity follows the Grothuss (hopping) mechanism that led to the formation of a network of H-bonds providing a pathway to H^+^, including interactions of N–H^+^ (imidazole)/H_2_PO_4_^-^, N–H^+^(pyridine)/H_2_PO_4_^−^, H_3_PO_4_/H_2_PO_4_^−^, H_3_PO_4_/H_2_O (in presence of a small portion of H_2_O). This explanation is in harmony with the proton transport mechanism report for PA-doped PBI under anhydrous and partially humidified conditions [20]. The values of *E_a_* of the present PA-doped membranes was found to be smaller than PBI membrane doped with PA of 14 M (11.7 kJ/mol) suggesting better proton mobility in the present membranes [56].

#### 3.5.8. Chemical Stability Changes

Figure 14 shows the weight loss (%) vs. time during rinsing of ETFE-*g*-P(4-VP/1-VIm)/PA membrane in the Fenton reagent. A tiny steady weight loss could be observed in the first 144 h, beyond which it slightly increased, reaching about 9% after 204 h of treatment with the Fenton reagent. This was accompanied by the absence of any physical deformation and hence it can be suggested that the present membrane has good chemical stability. The chemical stability of such a membrane was attributed to the formation of pyridine-N-oxide by the reaction of nitrogen atoms originating from the pyridine unit with hydroxyl radicals in the solution. Thus, the oxidation reactions of imidazole groups are hampered leading to high chemical resistance in the membranes [36]. A similar chemical stability trend was recently reported for the poly(pyridobisimidazole) membrane with enhanced acid doping under similar Fenton test treatment conditions [40].

#### 3.5.9. Summary of Membrane Properties

To realize the significant improvements introduced to the membrane obtained by acid doping of basic grafted ETFE grafted by a mixture of 4-VP/1-VIm, the properties of ETFE-*g*-P(4-VP/1-VIm)/PA and ETFE-*g*-P(4-VP)/PA were compared to PBI/PA membranes obtained from the literature [29,34] as shown in Table 2. The ETFE-*g*-P(4-VP/1-VIm)/PA membrane has a combination of excellent physical and chemical properties. For instance, it has acid DL, proton conductivity, and tensile strength superior to PBI/PA. Hence, this membrane is promising and can be suggested for further development.

### 3.6. Performance in HT-PEMFC

Figure 15 shows the performance of the membranes in terms of polarization and power density curves of ETFE-*g*-P(4-VP-*co*-1-VIm)/PA and ETFE-*g*-P(4-VP)/PA membranes in HT-PEMFC single cell under anhydrous and 20% RH conditions. The cell performance under both anhydrous and 20% RH conditions was considerably higher for ETFE-*g*-P(4-VP/1-VIm)/PA than ETFE-*g*-P(4-VP)/PA at 120 °C. For instance, the cell displayed an OCV of 970 mV under anhydrous conditions, which increased to 980 mV under 20% RH for ETFE-g-P(4-VP/1-VIm)/PA membrane compared to 960 and 950 mV for ETFE-*g*-P(4-VP)/PA under anhydrous and 20% RH conditions, respectively. Moreover, ETFE-g-P(4-VP/1-VIm)/PA showed a maximum power density of 226 mW/cm^2^ at anhydrous conditions that was increased to 278 mW/cm^2^ at 20% RH. Meanwhile, ETFE-*g*-P(4-VP)/PA demonstrated a lower maximum power density of 131 mW/cm^2^ at anhydrous conditions that was increased to 147 mW/cm^2^ at 20% RH. These values confirm the superiority of ETFE-*g*-P(4-VP/1-VIm)/PA membrane performance compared to ETFE-*g*-P(4-VP)/PA. The power peak value of ETFE-*g*-P(4-VP/1-VIm)/PA membrane (at 20% RH) was found to be remarkably higher than that of similar membranes recently reported in the literature tested at 110 °C and 20% RH [35]. The high performance of the partially humidified membrane can be attributed to the presence of water molecules that solvated the protons facilitating their mobility between the neighboring conductive sites involving H_2_PO_4_^-^ anion and protonated pyridine/imidazole rings. This enhanced the proton exchange activity (by hopping) through forming H-bond network. Nevertheless, ETFE-*g*-P(4-VP-*co*-1-VIm)/PA membrane performance under anhydrous conditions seems to be promising for HT-PEMFC and promotes further development.

## 4. Conclusions

Preparation of PA-doped PCM for HT-PEMFC was successfully carried out using RIG of a binary mixture of 4-VP and 1-VIm onto ETFE film. The grafting of such a pair of vinyl heterocyclic monomers allowed higher PA acid-doping levels than counterparts prepared by grafting of single monomers. The variation of grafting parameters allowed control of DG to desired levels suitable for fuel-cell application. DG of 59% was imparted to the grafted film at absorbed dose, monomer concentration, temperature, and reaction time of 100 kGy, 60 vol% (50:50 *v/v*: 4-VP and 1-VIm), 60 °C, and 18 h, respectively. The monomer reactivity ratios affected the composition of graft copolymer film and were found to be 0.78 for 4-VP compared to 0.23 for 1-VIm. The product of the reactivity ratio was found to equal 0.29, which is below the unity, proving that there is a tendency to deviate from the ideal alternation of comonomer units in the graft copolymer. The maximum value of the run number was obtained at a molar fraction of 0.6 4-VP at which the copolymer chains started to alternate, suggesting the dominance of 4-VP units in the grafted film. The DL of PA was a function of DG, PA concentration, temperature, and treatment time, and reached a maximum value of 119% for the sample having 59% DG, 85% PA, 80 °C temperature, and 4 days, respectively. The membrane showed homogenous graft distribution combined with appealing physicochemical properties in addition to a good chemical stability. The proton conductivity reached a value of 75.4 mS/cm at 120 °C under anhydrous conditions, which is about 10-fold higher than that of the PA-PBI membrane at 120 °C. The fuel cell test demonstrated a maximum power density of 278 mw/cm^2^ at 20% RH compared to 226 mw/cm^2^ at anhydrous conditions. The overall results suggest that the ETFE-*g*-P(4-VP/1-VIm)/PA membrane is a promising candidate for application in HT-PEMFC, and will stimulate more research.

## Figures and Tables

**Figure 1 membranes-13-00105-f001:**
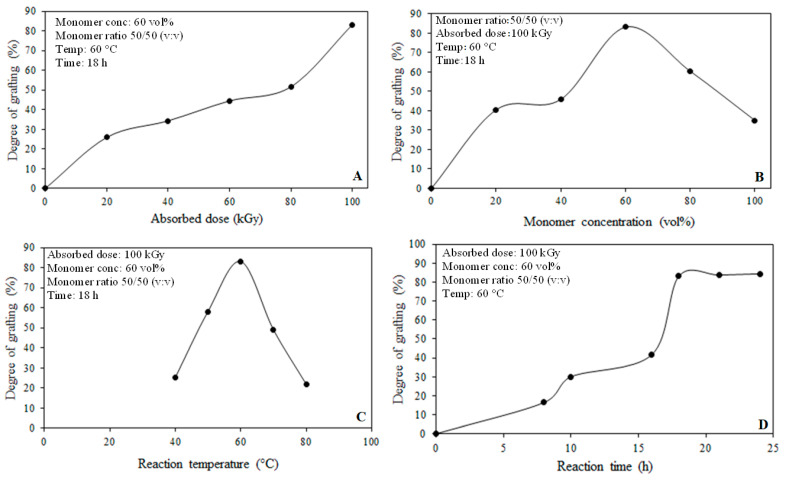
Variation of degree of grafting with reaction parameters: (**A**) absorbed dose, (**B**) monomer concentration, (**C**) reaction temperature, and (**D**) reaction time.

**Figure 2 membranes-13-00105-f002:**
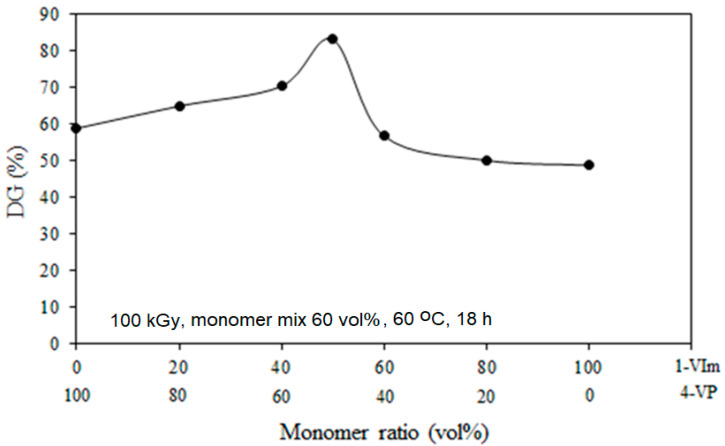
Variation of the degree of grafting with monomer ratio for grafting mixtures of 4-VP/1-VIm onto ETFE films.

**Figure 3 membranes-13-00105-f003:**
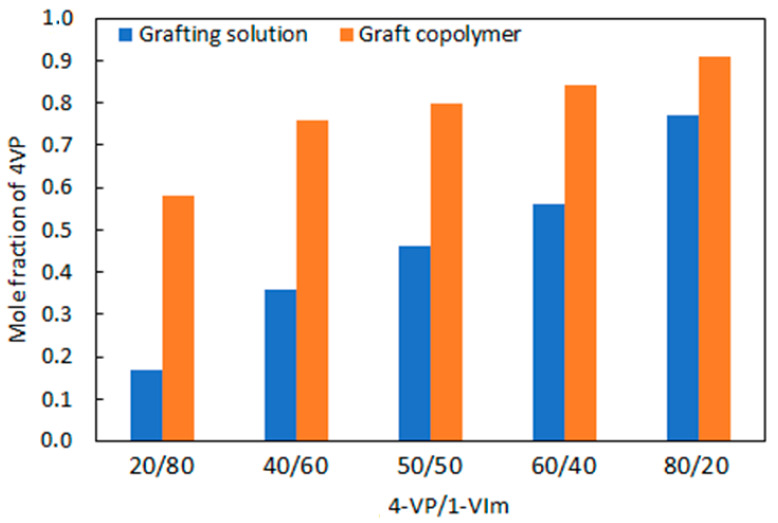
Variation of mole fraction of 4-VP in grafting mixture (blue) and in graft copolymer (orange). Grafting parameters: monomer conc. of 60 vol%, absorbed dose of 100 kGy, reaction time of 18 h, and temperature of 60 °C.

**Figure 4 membranes-13-00105-f004:**
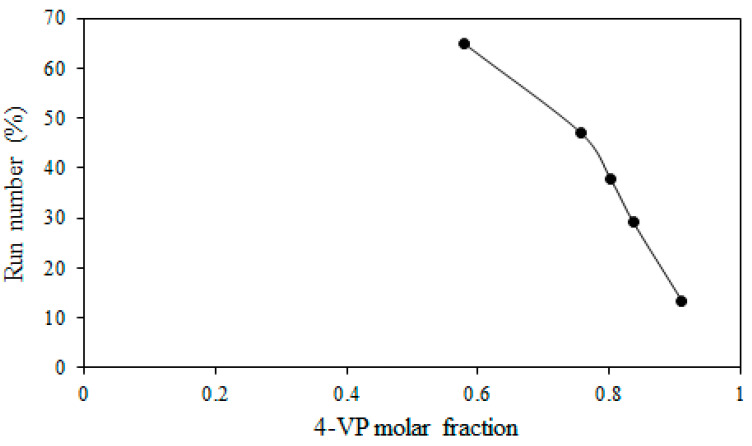
Variation of run number with 4-VP mole fraction in grafting of 4-VP/1-VIm mixtures onto ETFE film.

**Figure 5 membranes-13-00105-f005:**
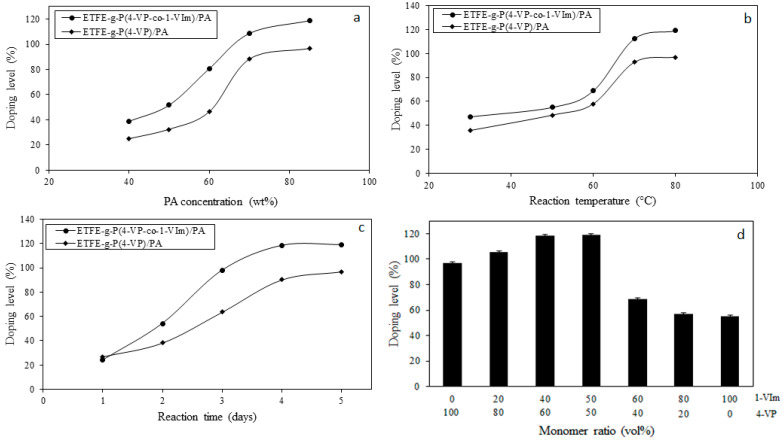
Variation of PA doping level of P(4-VP/1-VIm) grafted films with reaction parameters: (**a**) PA concentration, (**b**) reaction temperature, (**c**) reaction time, and (**d**) monomer ratio in grafting solution.

**Figure 6 membranes-13-00105-f006:**
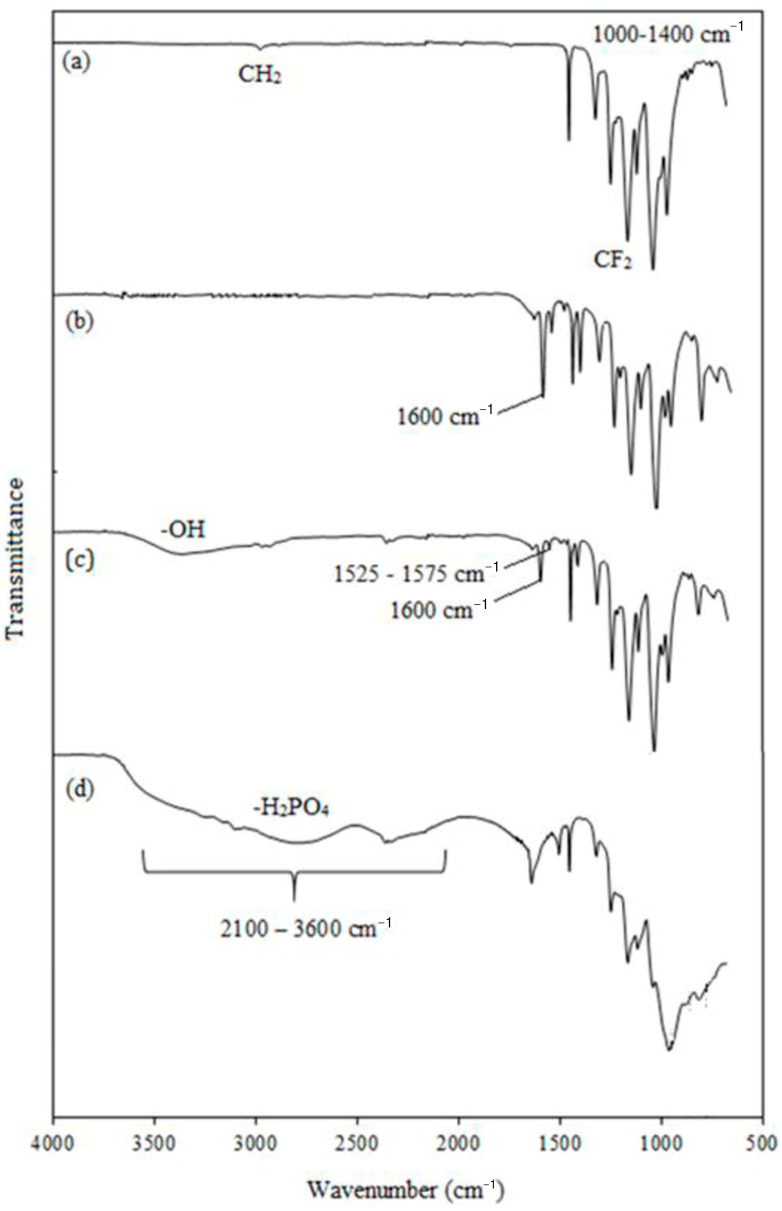
FTIR spectra of: (**a**) pristine ETFE film, (**b**) ETFE-*g*-P(4-VP) film, (**c**) ETFE-*g*-P(4-VP/1-VIm) film, and (**d**) ETFE-*g*-P(4-VP/1-VIm)/PA membrane with 59% DG.

**Figure 7 membranes-13-00105-f007:**
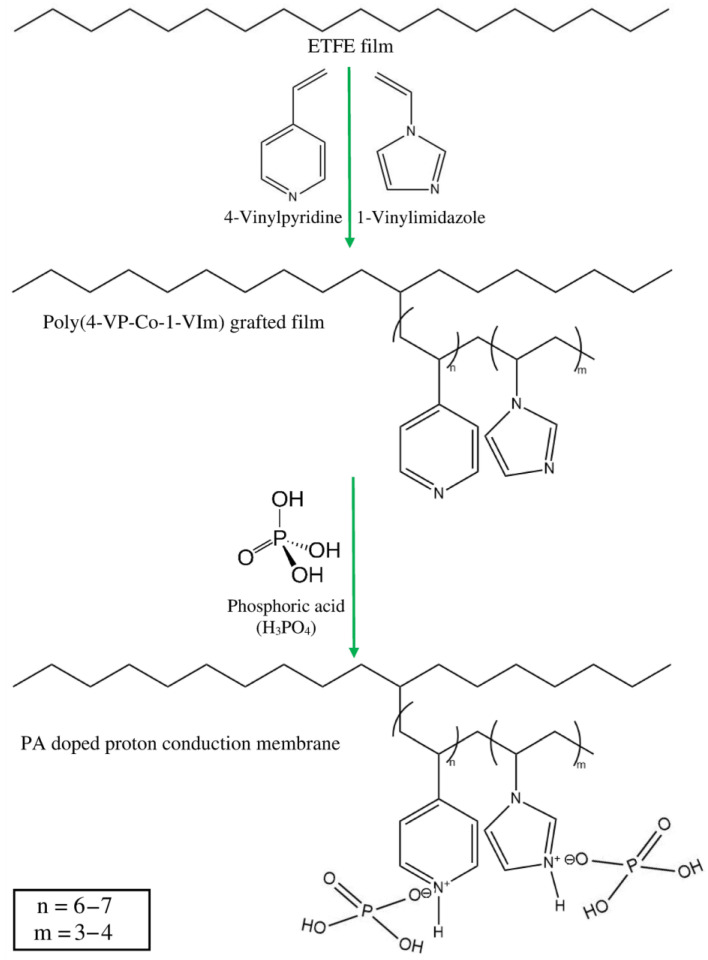
Schematic representation of preparation of PA-doped membrane obtained by RIGC of 4-VP/1-VIm mixture onto ETFE film.

**Figure 8 membranes-13-00105-f008:**
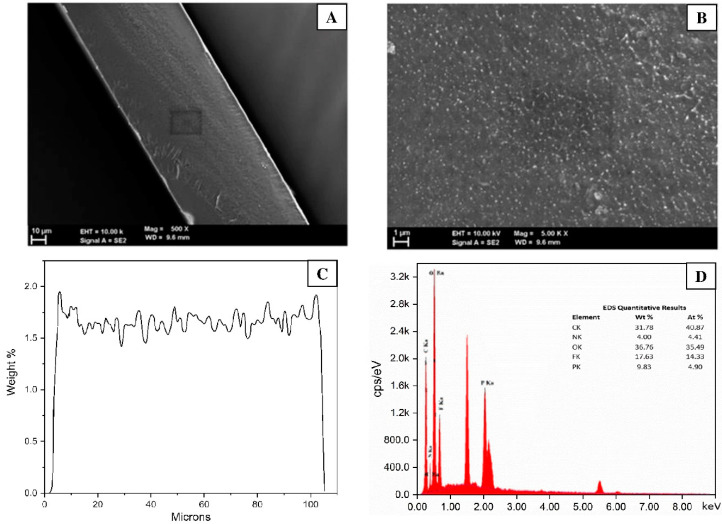
Cross-sectional view of ETFE-*g*-P(4-VP/1-VIm)/PA membrane with: (**A**) low magnification, (**B**) high magnification and (**C**) phosphorus mapping across membrane together with (**D**) EDX image of membrane elemental composition.

**Figure 9 membranes-13-00105-f009:**
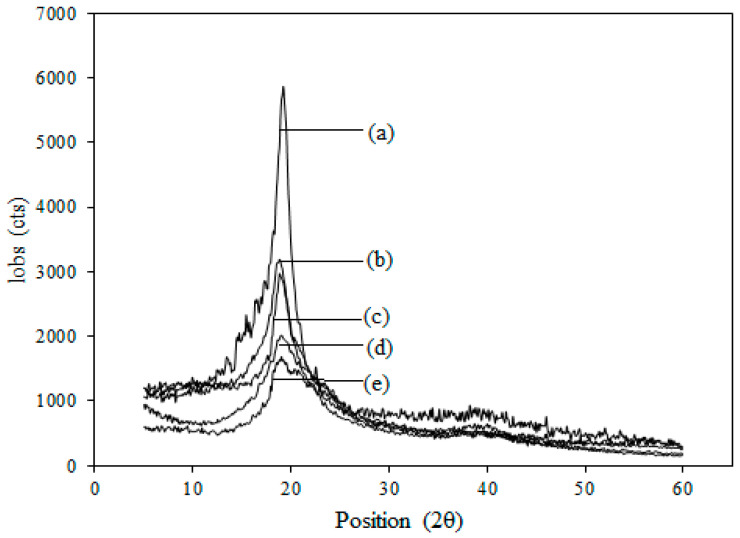
XRD diffractograms of: (a) pristine ETFE film, (b) ETFE-*g*-P(4-VP)-grafted film, (c) ETFE-*g*-P(4-VP/1-VIm)-grafted film, (d) acid-doped P(4-VP) membrane and (e) acid-doped P(4-VP/1-VIm) membrane.

**Figure 10 membranes-13-00105-f010:**
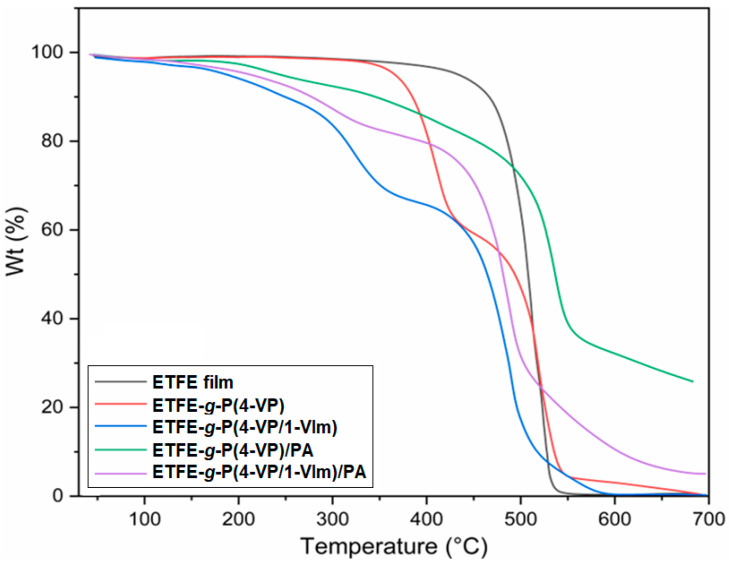
TGA thermograms of pristine ETFE film, ETFE-*g*-P(4-VP), and ETFE-*g*-P(4-VP/1-VIm) films compared to ETFE-*g*-P(4-VP)/PA and ETFE-*g*-P(4-VP/1-VIm)/PA membranes.

**Figure 11 membranes-13-00105-f011:**
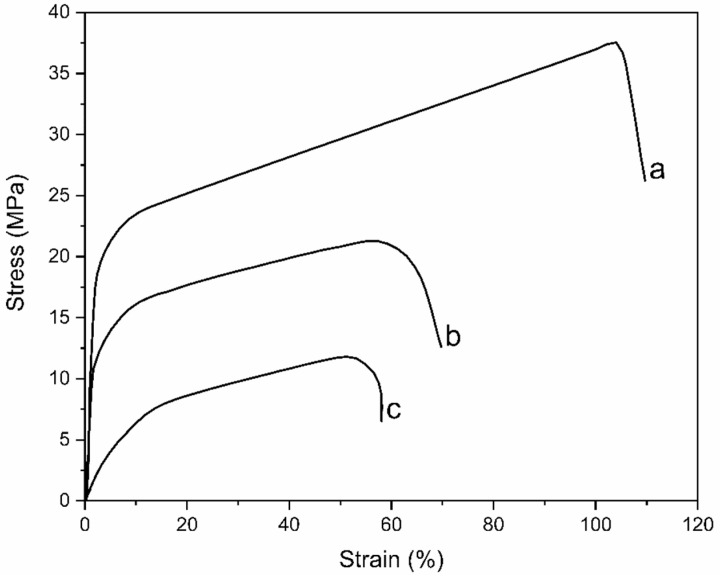
Stress-strain curves of: (a) pristine ETFE, (b) ETFE-*g*-P(4-VP/1-VIm)-grafted film, and (c) PA doped-membrane.

**Figure 12 membranes-13-00105-f012:**
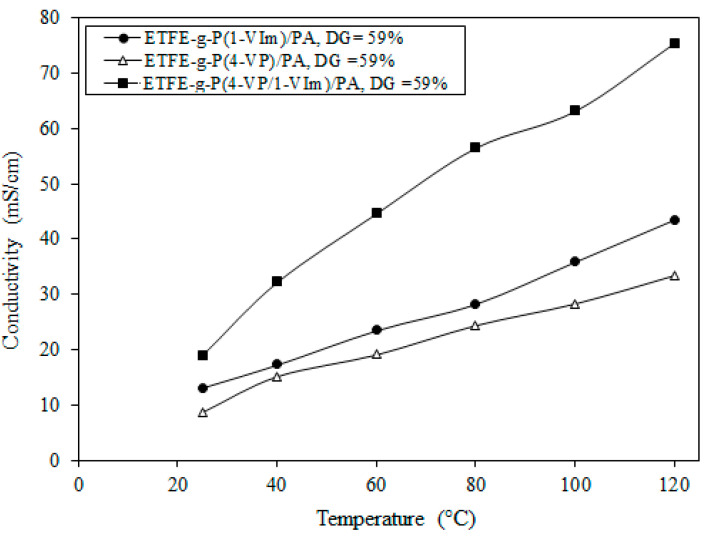
Variation of proton conductivity with temperature at anhydrous conditions for ETFE-*g*-P(4-VP/1-VIm)/PA compared to membranes obtained from the grafting of individual monomers.

**Figure 13 membranes-13-00105-f013:**
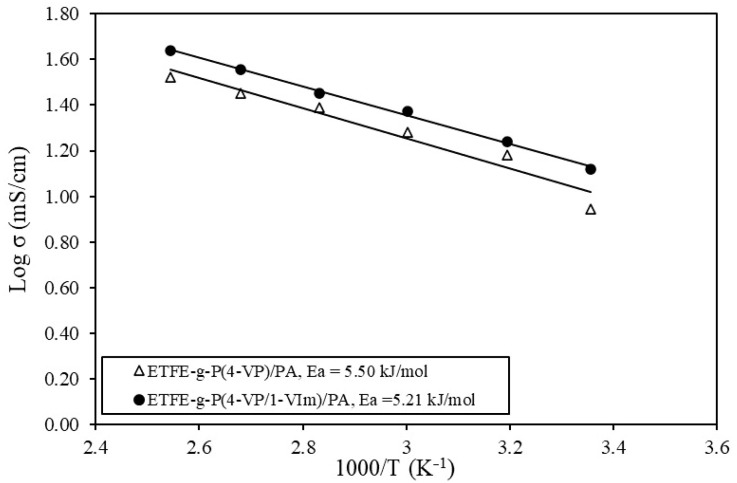
Arrhenius plots for proton conductivity vs. reciprocal of temperature for PA doped membranes.

**Figure 14 membranes-13-00105-f014:**
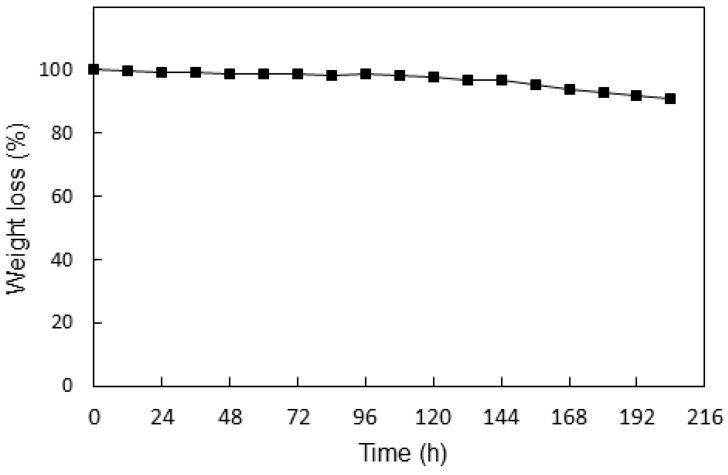
Variation of weight loss (%) with time during treatment of ETFE-g-P(4-VP/1-VIm)/PA membrane in Fenton reagent.

**Figure 15 membranes-13-00105-f015:**
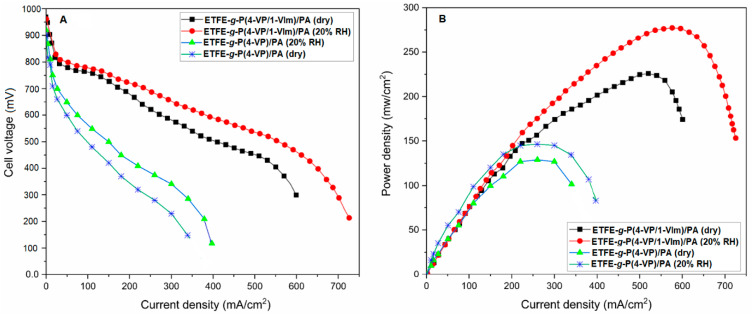
Performance of HT-PEMFC single cell with PA doped membranes (DG = 59% with a doping level of 119%): (**A**) polarization curves and (**B**) power density curves, tested under an anhydrous condition and 20% RH. H_2_ flow rate is 450 mL/min and that of O_2_ is 300 mL/min at temp. of 120 °C.

**Table 1 membranes-13-00105-t001:** TGA data of various degradation transitions of pristine ETFE film, ETFE-*g*-P(4-VP), and ETFE-*g*-P(4-VP/1-VIm) films compared to ETFE-*g*-P(4-VP)/PA and ETFE-*g*-P(4-VP/1-VIm)/PA membranes.

Sample	Dehydration	PA Evaporation	Depolymerization	ETFE Degradation
ETFE film	-	-	-	440 °C
ETFE-*g*-P(4-VP)	-	-	340 °C	470 °C
ETFE-*g*-P(4-VP/1-Vm)	150 °C	-	290 °C	445 °C
ETFE-*g*-P(4-VP)/PA	150 °C	200 °C	340 °C	485 °C
ETFE-*g*-P(4-VP/1-Vm)/PA	150 °C	200 °C	320 °C	450 °C

**Table 2 membranes-13-00105-t002:** Summary of the properties of ETFE-*g*-P(4-VP/1-VIm)/PA and ETFE-*g*-P(4-VP)/PA membranes.

Properties	P(4VP/VIm)/PA	P(4-VP)/PA	PBI/PA [29,34]
DG (%)	59	59	N/A
DL (%)	119	97	61
Thickness (µm)	105	104	100
Thermal stability (°C)	200	200	200
Proton conductivity at 120 °C (mS/cm)	75.4	33.4	9.6
Tensile strength (MPa)	12.3	17	3.4

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
