# Peer review of "Composite Proton-Conducting Membrane with Enhanced Phosphoric Acid Doping of Basic Films Radiochemically Grafted with Binary Vinyl Heterocyclic Monomer Mixtures"

_membranes, 2023, doi:10.3390/membranes13010105_

Round 1

Reviewer 1 Report

The manuscript reported the composite PEM by radiation-induced grafting based on ETFE film followed by PA doping for HT-PEMFC applications. The various grafting parameters were varied to control the degree of grafting. The mechanical properties and proton conductivity were evaluated, and the PEMFC performance was tested at 120oC. It was found that the acid doping level was enhanced by increasing the degree of drafting. The obtained composite membranes also showed great potential for future applications in HT-PEMFCs.

I consider the content of this manuscript meets the reading interests of the readers of the journal. However, there are certain English spelling and grammar issues, and also the discussion and explanation should be further improved. I suggest giving a minor revision and the authors need to clarify some issues or supply more experimental data to enrich the content. This could be comprehensive and meaningful work after revision.

1. For grammar issues, it is suggested that the author double-check the small grammar errors in the full text, especially the lack of and redundant use of definite articles.

2. For the Keywords, chemical and physical property, proton conductivity, and fuel cell performance should be added in order to attract a broader readership.

3. Page 1, ‘The interest in the development of composite proton-conducting membranes (PCMs) with high conductivity and durability for high-temperature proton exchange membrane fuel cells (HT-PEMFC) is receiving increasing attention [1]. 

The direct appearance of HT-PEMFC at the beginning is too abrupt. Readers may wonder why HT and LT are required at the very beginning. Also, what is a fuel cell? What are the application scenario and era background of fuel cells? I suggest there should be a brief introduction to FC at the beginning. Especially in the context of the global promotion of decarbonization, green hydrogen energy is favoured. And compared with renewable energy (such as wind energy and solar energy), which are unstable and intermittent during generation, fuel cell techniques based on hydrogen are much simple since there is no need to employ additional employment of energy storage systems to improve the utilization rate and stability (ChemSusChem, 2022, 15(1): e202101798).

This is due to the advantages of HT-PEMFC including high CO tolerance, enhanced cathode kinetics, and reduced system complexity resulting from the elimination of the water management system [2]. All the advantages should be compared with LT-PEMFC, and this should be clarified. Otherwise, compared to SOFC or AFC, HT-PEMFC may not possess these advantages anymore.

4. Page 4, there is nothing about the basic physicochemical parameters measured for the obtained membranes. For example, water uptake, swelling ratio, ion exchange capacity, contact angle, and so on. I suggest adding these results, see Table 1 (Journal of Membrane Science, 2017, 527: 35-42).

  For the TGA measurement, how about the flux of the nitrogen gas flow?

5. Page 13 and 14, for the TGA, the various thermal events cannot be distinguished very clearly. Hence, I suggest adding the first derivative of the wt% ascribed to mass loss to make the results more easily read. An example can be found in Electrochimica Acta, 2019, 309: 311-325, Figure 1.

6. The resolution of Figures 8c and 8d is too low, which should be modified to a higher resolution one.

7. For Table 1, it is good to compare the obtained membranes with PBI/PA. But in the chemical stability part, only one sample is shown, with no comparisons with other membranes, or even PBI/PA. For Figure 12 proton conductivity, the figure should also include the PBI membrane. The same applies to the fuel cell test, the comparisons with PBI/PA membrane are also important to highlight the importance of this work.

Author Response

I would like to thank the reviewer for the invaluable comments that have been carefully dealt with, and appropriate responses have been made as shown in the attached rebuttal reports. The red color writing is for the required amendments and the blue color is for the additional modifications and improvements.

Reviewer 2 Report

Composite membranes were prepared by RIG of 4-VP and 1-VIm onto ETFE film for HT-PEMFC applications, followed by PA doping. The changes in the chemical and physical properties endowed by grafting and subsequent PA acid doping were investigated. It is found that the acid doping level was enhanced by increasing DG. The obtained membranes were found to have superior properties compared to counterparts obtained from the grafting of single monomers. The contents are interesting and make sense. I give some suggestions for certain possible revisions.

Detailed comments can be found in the PDF file.

Author Response

(The authors gave the same response as above.)
